# Research on an Adaptive Real-Time Scheduling Method of Dynamic Job-Shop Based on Reinforcement Learning

Haihua Zhu *[ID], Shuai Tao, Yong Gui [ID] and Qixiang Cai

College of Mechanical and Electrical Engineering, Nanjing University of Aeronautics and Astronautics, Nanjing 210016, China
* Correspondence: h.zhu@nuaa.edu.cn

**Abstract:** With the rapid development of modern industrialization in our country and the continuous improvement of people's living standards, the changing market has put forward new requirements for traditional manufacturing enterprises. The order product demand of many manufacturing enterprises is changing from a single variety, large batch to multiple varieties, small batch. In view of this change, the traditional job-shop scheduling method is far from enough, which greatly affects the efficiency of the production job-shop. In order to solve the above problems, this paper proposes a real-time scheduling method based on reinforcement learning applied in the dynamic job-shop and a new type of neural network is designed at the same time. The neural network is designed with the high-dimensional data in the above problem as input, and a policy-based reinforcement learning algorithm is proposed based on this network. In the process of research, it was found that the reinforcement learning method not only enables the agent to use historical data for learning, but also enables it to explore and learn other possible high-reward actions within a certain range, so as to realize the optimization of production goals under real-time scheduling. The effectiveness of the proposed real-time scheduling method is verified by comparing with other common rule-based scheduling methods in the manufacturing environment.

**Keywords:** real-time scheduling method; dynamic job-shop; reinforcement learning; policy-based

## 1. Introduction

With the continuous advancement of the industrialization process and the improvement of people's living standards, people have begun to pursue the uniqueness of products under the premise of quality assurance of product. The traditional production mode of single variety and large scale is gradually changing to multiple types and small batch production to meet the customization requirements [1]. However, this transition has also brought many unavoidable phenomena, such as the increase in product types, the decrease in production volume and unstable demand. These uncertain factors are reflected in the changes in the processing links of products, the increase in mechanical equipment in the job-shop and the changes in the types of orders of enterprises, which seriously affect the production efficiency [2]. Therefore, it is urgent to deal with these changes in dynamic job-shop in real time and improve the efficiency of job-shop production scheduling [3].

As an important technical means to improve the utilization of manufacturing resources and realize efficient production operation, production scheduling is the core of production process control. The optimized scheduling scheme can greatly improve the performance of manufacturing system, reduce costs and improve productivity [4]. In the literature, there are a variety of methods for production scheduling. Mathematical programming approaches achieve global optimal schedules by obtaining critical solutions of programming functions under constraints. Metaheuristic algorithms, such as genetic algorithm (GA) [5] and ant colony algorithm [6], are introduced to simulate manufacturing systems with simple and natural rules. The advantage of the above methods is that they can get obtain optimal or

near-optimal solution. However, they are traditionally implemented offline in a centralized manner with static and deterministic assumptions and have poor real-time performance in the dynamic manufacturing environment [7]. Dynamic scheduling is aimed at dealing with real-time changes, but its rules are man-made, which are fixed and limit the action selections [8]. Hence, in order to meet current manufacturing systems' requirement, it is in an urgent need to propose a new intelligent scheduling method.

With the rapid development of emerging technologies, especially new computer technologies such as Internet of things and artificial intelligence (AI), it offers the possibility of increasing information visibility about real-time statuses of manufacturing resources. More and more real-time manufacturing data can be acquired automatically [9]. Large amounts of data are readily available in the manufacturing system, which provides an unprecedented opportunity to improve the performance of the manufacturing system. Recently, AI fuels an increasing interest to solve dynamic scheduling problems by continuously improving the scheduling performance of schedulers [10]. AI methods provide attractive features such as dynamic adjustment, online learning, and real-time decision making. This provides an unprecedented opportunity to handle unexpected events in real time and improves the decision-making abilities of schedulers in the manufacturing processes. Reinforcement learning (RL) is a rapid, efficient and irreplaceable learning algorithm, which belongs to a branch of AI. In recent years, it has been studied and applied in the process of job-shop production scheduling by more and more people, which has a good effect in improving the production efficiency of the job-shop [11,12]. Although most RL-based studies improve scheduling efficiency at different levels, the essence of them was to let agents learn historical data and experience, instead of interactive learning and rescheduling according to the real-time information change in dynamic job-shop.

According to the above, in order to improve the real-time interactive learning and dynamic adjustable ability in the process of job-shop production scheduling, a real-time scheduling method of dynamic flexible job-shop based on reinforcement learning was proposed. Based on the study of Markov Decision Process (MDP), a model of MDP for Flexible Job-shop Scheduling Problem(FJSP) is established by using the similarities and relationships between the MDP and the shop scheduling model [13]. Then, a learning neural network based on the state-action value of production scheduling is designed. In the end, a policy-based deep reinforcement learning algorithm is proposed to achieve the optimization of objectives. The rest of the paper is organized as follows. Section 2 contains some related research status of static or dynamic job-shop production scheduling methods. Section 3 introduces the main research methodology. Experiment and results are given in Section 4. Finally, conclusions are shown in Section 5.

## 2. Research Background

In order to solve the real-time scheduling problem of dynamic job-shop, this paper studies the technology and algorithm of job-shop scheduling and focuses on the static production scheduling and intelligent production scheduling methods of job-shop.

### 2.1. Static Production Scheduling Methods

Shop scheduling problem is a classic NP-hard problem, which is extremely challenging and has attracted the attention of many scholars. At present, most of the existing solutions to the flexible job-shop scheduling problem assume that the production environment is static. Wu [14] used the improved particle swarm optimization algorithm as the solution method to solve the flexible job-shop scheduling problem. Tian et al. [15] proposed an improved Gray Wolf algorithm based on the classical Gray Wolf algorithm to optimize the maximum completion time to solve the flexible job-shop scheduling problem. Dong [16] proposed a hybrid genetic and ant colony algorithm for flexible job-shop scheduling problem. Sha [17] modified some Settings in PSO to enable multi-objective optimization. Zhou [18] studied the artificial bee colony algorithm for flexible job-shop scheduling problem. Li [19] designed a flexible job-shop batch scheduling algorithm by using the framework of genetic algorithm

and a flexible batch coding method based on "cursors". Cheng [20,21] et al. proposed an improved hybrid genetic algorithm to solve the flexible job-shop scheduling problem.

However, the above research aims at solving the static scheduling problem, which is obtained under the premise that all factors do not change. But in the actual production environment, real-time scheduling is often required according to the current situation of the job-shop [22], so an intelligent scheduling method is needed to solve this problem.

### 2.2. Intelligent Production Scheduling Methods

In order to solve the dynamic job-shop scheduling problem, many intelligent scheduling methods are used, such as rule-based scheduling method, simulation scheduling method, artificial intelligence scheduling method and so on. Lv [23,24] proposed the MAS interaction mechanism of combined auction and achieved good results in agile production scheduling. This kind of method has low computation cost and global properties, but it cannot realize knowledge learning and acquisition. Cai et al. [25] developed a large-scale and dynamic job-shop simulation platform. Zhang et al. [26] proposed a multi-intelligence architecture system integrating self-organizing negotiation and self-learning strategies.

Although all the above methods have high solution quality, they may take a long time and are not suitable for real-time scheduling of dynamic job-shop. In recent years, reinforcement learning (RL) in the field of artificial intelligence has become an effective method to solve this problem. Many research methods based on RL have been applied to dynamic scheduling of different kinds of job-shops. Chang et al. [27] proposed deep reinforcement learning (DRL) to solve the dynamic FJSP (DFJSP) with random job arrival, with the goal of minimizing penalties for earliness and tardiness. Johnson [28] proposed a Multi-Agent Reinforcement Learning (MARL) system for scheduling dynamically arriving assembly jobs in a robot assembly cell. Zhou et al. [13] presented new cyber-physical integration in smart factories for online scheduling of low-volume-high-mix orders and the design of new reward functions to improve the decision-making abilities of multiple AI schedulers based on reinforcement learning (RL). Waschneck et al. [29] designed some cooperative Deep Q-Network(DQN)-based agents for a job-shop, where each agent optimizes the dispatching rules at one work center while monitoring the actions of other agents and optimizing a global reward. Khalil et al. [30] proposed a fitted Q-learning based on a deep learning architecture over graphs to learn greedy policies for a diverse range of combinatorial optimization problems. Han et al. [31] proposed an end-to-end deep reinforcement learning (DRL) framework based on 3D disjunctive graph dispatching. Aldarmi, SA et al. [32] proposed value-based scheduling algorithms to improve the performance under normal load in the scheduling system and to reduce the performance degradation of the system under overload to a certain extent. Tseng, SM et al. [33] present an adaptive real-time scheduling policy for scheduling value-based transactions in a multiprocessor real-time database system. Prasad, D [34] proposed a valued-based scheduling method to select the optimal service operation autonomously. Compared with the traditional neural network, the strategy network of the new manufacturing neural network proposed in this paper is composed of several assignment action policy networks and an execution action policy network. Each action assignment will be completed by a policy network and cooperate with the execution action policy network to make the action performed more reasonable.

In this paper, the reinforcement learning model of flexible job-shop scheduling is established with a manufacturing neural network, and the training method of the scheduling agent is studied based on this model.

## 3. Research Methodology

At present, in most production job-shops, jobs are transported from the warehouse to the production job-shop by an AGV (Automatic Guided Vehicle) trolley and other transportation equipment, and then the robotic arm and transportation equipment cooperate with each other to move these jobs to the designated processing machine for processing according

to the instructions. The overall completion process of the job order and corresponding process of the proposed adaptive real-time scheduling method is shown in Figure 1.

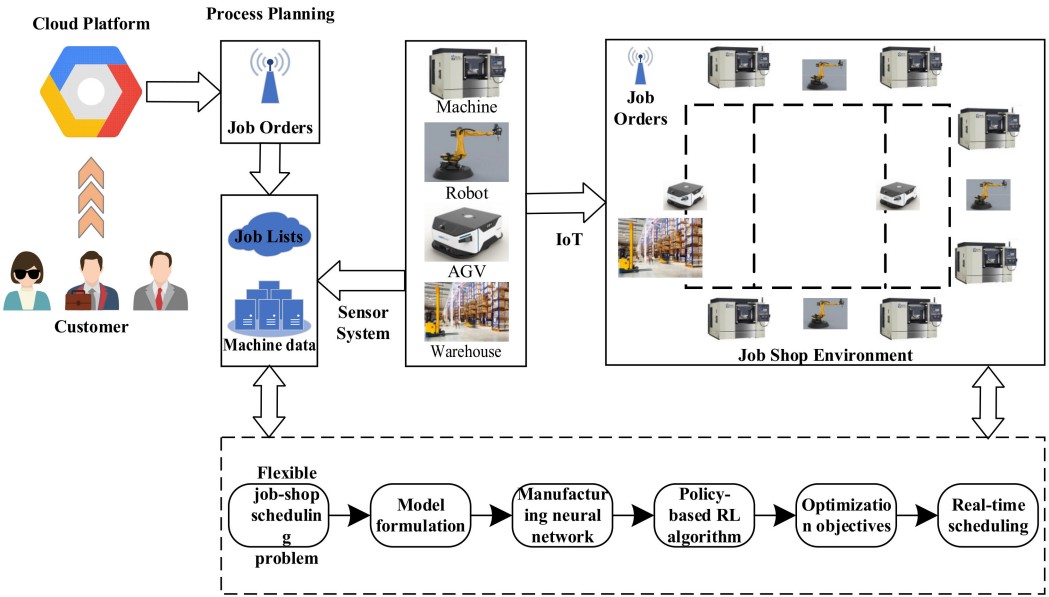

**Figure 1.** Flowchart of the proposed methodology.

Firstly, the orders for personalized products are submitted by customers on the cloud platform, and then the information of these orders are sent to the database on the production floor. Then job-shop production system decomposes the order data into a series of manufacturing operation information, and the working state of the machine and the corresponding operation information of jobs are obtained in real time by advanced sensors [14]. Next, the scheduling problems will be described by the mathematical model and the designed corresponding neural network is used to generate a scheduling policy. The policy-based algorithm is applied to the neural network for training, so that the objective optimization is achieved to realize the scheduling of the production job-shop [28].

### 3.1. Mathematical Model

#### 3.1.1. The Mathematical Model of FJSP

In a FJSP, there exist several machines and several jobs. NC represents the total number of jobs; NM represents the total number of machines; A machine is denoted by $Mm$, $m = 1, \ldots , NM$; a job is denoted by $Ji$, $i = 1, \ldots ,$ NC; each job consists of a series of operations with sequence constraints. The $j_{th}$ operation of a job Ji is denoted by $Oij$, $j = 1, \ldots , Ji$; when an operation is completed, the next operation of the same job will be initialized and appended to the scheduling list. If there is no available machine for the current operation, it will wait for the next scheduling moment. If two or more operations are initialized at the same time, the scheduler will prioritize the one with more rewards.

For the convenience of describing, here are some assumptions: First, each machine can only process one specific type of operation (e.g., turning, milling, or drilling), and each job can only be processed on one machine at one time; each operation can be completed in one machine; machines cannot be interrupted when processing a job; transporting time of job-shops is ignored, i.e., every operation can immediately be processed whenever its previous one has been carried out; machines' setup time is ignored; machines of the same specifications are totally the same; and for the convenience of designing the objective function, we focus on a system's makespan rather than a jobs' constraints of delivery.

### 3.1.2. The Principle of Applying RL to Model FJSP

Reinforcement learning based on the Markov decision process is a way of on-line learning which can be applied to a single agent environment. MDP is a serialized model. In each trail, an agent repeatedly reads current state and chooses and implements an action. Meanwhile, the environment repeatedly transfers from one state to another and returns a reward until to the end state. The process of MDPs is shown in Figure 2.

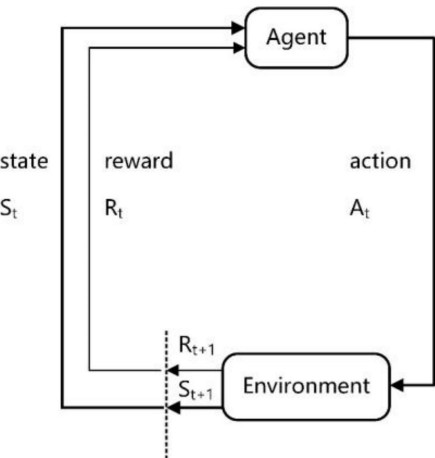

**Figure 2.** The process of MDPs.

The job-shop production scheduling problem is also a serialized decision problem in nature and can also be described in the framework described above. In the production job-shop scheduling system, when a certain operation of one job is completed, the next operation of this job will be added to the list waiting for scheduled. Then the scheduling system will make a decision to choose the next scheduling action according to the information feedback of the current job-shop environment and the job-shop production environment will be transformed to the next state. The corresponding relationships between MDP and FJSP are listed in Table 1.

**Table 1.** The connection between MDP and FJSP.

| MDP | FJSP |
|---|---|
| Agent | Scheduler |
| Environment | Production job-shop system |
| Trail | Scheduling route of single order |
| Action node | Scheduling moment |
| State | Dynamic properties of the job-shop environment |
| Action | Assigning the operation to the corresponding machine |
| State transition Reward | Completion time or other goals |

### 3.2. Manufacturing Neural Network

When there are some tasks to be scheduled, the features of the job-shop need to be fed to the manufacturing neural network to generate scheduling policies. The features of the job-shop environment (i.e., the input of the neural network model) consist of the process information of the jobs to be scheduled and the working state information of the machine tool. Specifically, the process information of the jobs to be scheduled includes the number of operations of every job, the arrival time of the jobs, and the working state information of the machine tool which includes the process time of every operation, etc. this information shows complexity and diversification, leading to the high dimension of the input of the neural network model.

To address the high dimensionality of jobs and machines, a manufacturing neural network is proposed to optimize dispatching rules. The neural network is divided into two parts: production behavior network and scheduling policy network. According to the job type, the scheduling policy network is also divided into multiples, in keeping with the number of job types. By inputting the specified type of job information and machine information, the scheduling policy network can decide whether the task should be assigned under the current environment characteristics and output the probability distribution of different alternative production actions. By inputting all the job information and machine information, the production behavior network can decide whether to continue processing under the current environmental characteristics. In summary, this type of neural network can determine whether the job-shop should process or schedule in the current environment based on the input feature information.

The detailed setup of network inputs, neural network, action and the reward are described as below:

(1) Scheduling moment

The moment when the scheduling system performs the scheduled action again after performing the previous action and being rewarded.

(2) Inputs of neural network

The current state of the job-shop $s_n$ is represented by some typical characteristic quantities of the environment. The scheduling information is made up of two types, as shown in Figure 3.

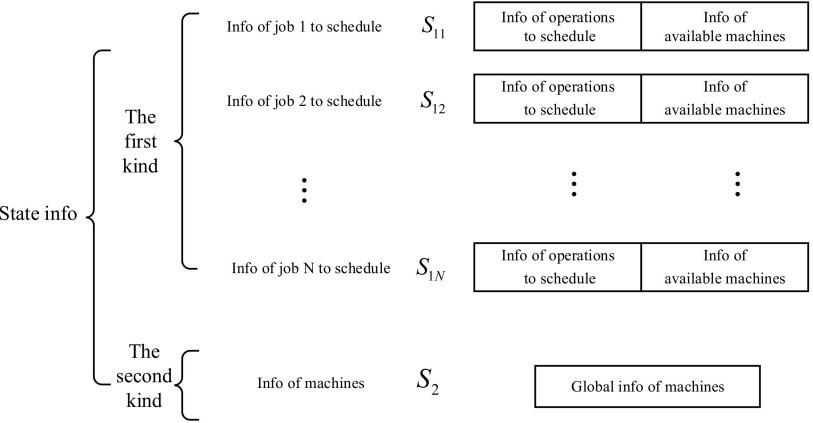

**Figure 3.** The information in the state.

The first type is the information of jobs to be scheduled, which includes the operation attributes of the job and the workload of available machines. It is denoted by $s_{1i}$, in this notation, where 1 indicates that the scheduling information belonging to the first type of state, and i indicates the local index of the job to be scheduled in the state space. In addition, $ind_i$ is used to represent the global index of the job.

The second type is the information of all machines in the job-shop, which contains the remaining processing time and the operations processing schedule of each machine. It is denoted by $s_2$, in this notation, where 2 indicates that the scheduling information belongs to the second kind of state.

By combining the above two kinds of scheduling information, the state space of job-shop can be obtained as $S_n = (s_{11}, s_{12}, \ldots s_{1P}, s_2)$.

(3) Framework of neural network

The role of the policy-based network is to generate a determined policy scheme by learning knowledge depending on a specific input. The overall structure of the neural network model is shown in Figure 4, which includes NT task allocation strategy networks and one machining strategy network.

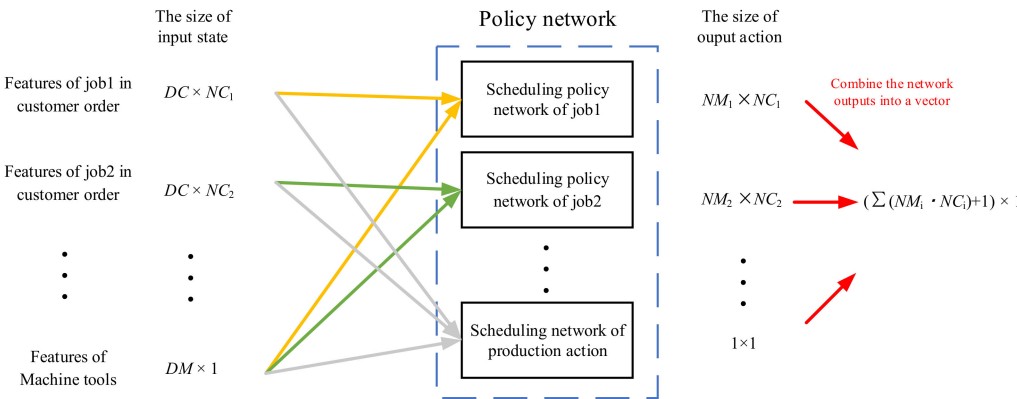

**Figure 4.** The structure of proposed neural network.

The input of the task allocation strategy network of job i is the job attributes and machine state information of the corresponding category. The output is the evaluation value of all optional allocation behavior of processing type i, and the size is $NM_i \times NC_i$. The input of the machining strategy network is the information of the job to be scheduled and the state information of all machines, the output value is the evaluation value of machining behavior, and the size of the data matrix is $1 \times 1$. Finally, all the outputs of the policy network are combined to obtain a behavior vector with the size of $1 \times (\sum(NC_i \cdot NM_i) + 1)$.

(4) Set of available actions

There are two kinds of action spaces in the set of available actions. The first category is assigning action, denoted by $a_{1ij}$. In this notation, 1 stands for the first class, i stands for the number of the selected job and j stands for the number of the selected machine. The second class is called manufacturing action, which indicates finishing the operations over time. It is denoted by $a_2$; 2 stands for the second kind.

The scheduling agent's available action space depends on the state where the environment is, denoted by $A_{Sn}$. In this notation, $S_n$ stands for the current state at the $n^{th}$ scheduling node. To determine whether an action is available, the formulas are listed as follows.

For a scheduling action $a_{1ij}$:

$$\begin{cases} a_{1ij} \in A_{sn} & j \in M_{ind_i} \\ a_{1ij} \notin A_{sn} & j \notin M_{ind_i} \end{cases}$$

$M_{ind_i}$ stands for a set of machines that have the ability to process the job $M_{ind_i}$.
For a processing action $a_{1ij}$:

$$\begin{cases} a_2 \in A_{sn} & M_{busy} = \emptyset \\ a_2 \notin A_{sn} & M_{busy} \neq \emptyset \end{cases}$$

$M_{busy}$ stands for a set of machines that have jobs to process, means an empty set.

(5) State transition

Two kinds of state transition will occur in this environment. The first class is the state transition of the job to be scheduled as shown in Figure 5. When an assigning action is executed, the assigned job will be removed from the jobs sequence to be scheduled; when a manufacturing action is executed, the next operation of the job will be added into the jobs sequence to be scheduled.

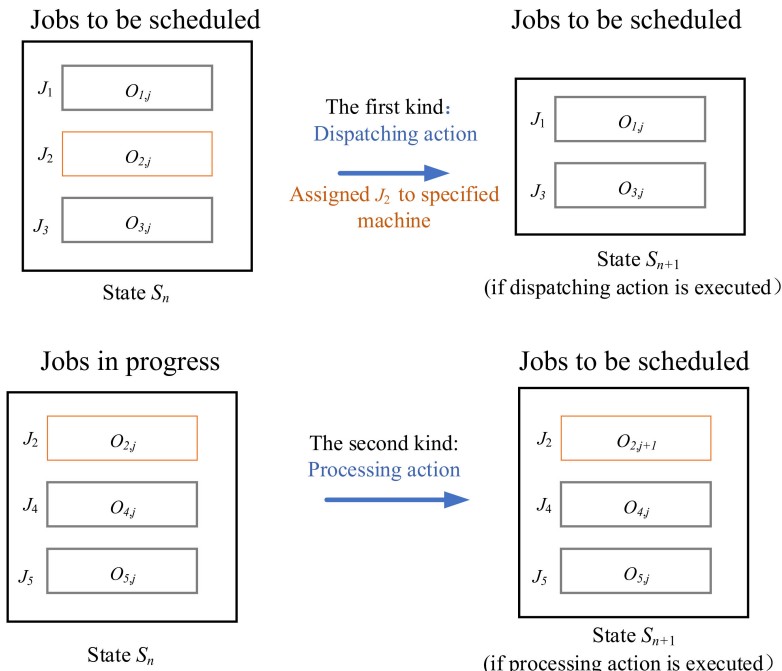

**Figure 5.** The state transition of the job.

The second kind is the state transition of the machine as shown in Figure 6. When an assigning action is executed, an operation will be assigned to a specified machine; when a processing action is executed, the state transition of the machine means that some operations have been finished.

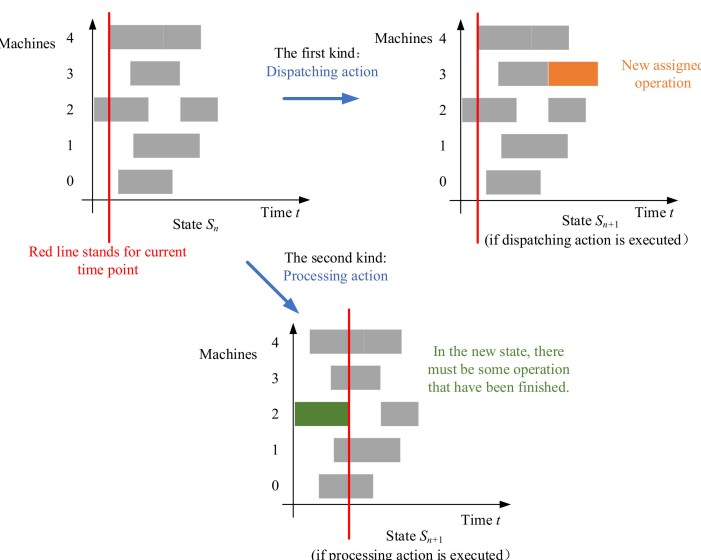

**Figure 6.** The state transition of the machine.

(6) Reward

In order to optimize the agent to achieve the goal, the reward is set to judge the positive and negative feedback of the executed action. Because the target in this article is to minimize the system's idle time, the reward is designed as follows.

$$r_{n+1} = \begin{cases} 0 & a_n \in A_1 \\ -F_n & a_n \in A_2 \end{cases}$$

In the equation, $r_{t+1}$ represents the reward; $A_1$ stands for the first kind of action, $A_2$ stands for the second kind; $F_n$ means the idle time between node $n$ and node $n + 1$.

### 3.3. The Learning Algorithm

After comparing the value-based algorithm and the policy-based algorithm, the latter is selected in this paper for the following reasons:

(1). The policy-based algorithm has a smoother exploring rule than the value-based algorithm.

(2). The policy-based algorithm has a better convergence property.

(3). The policy-based algorithm is effective in high-dimension problems and can learn stochastic policies.

#### 3.3.1. Policy Representation

In the policy-based algorithm, probability distribution is used to represent the agent's policy. $\pi(a|s, \theta)$ is the notation of the probability that the agent chooses action $a$ in state $s$; $\theta$ stands for the parameters of the approximator.

At any node, a priority $prefer_{a|s,\theta}$ is computed by NN for each action a. In the computation of priority, Gibbs distribution is used.

$$\pi(a|s, \theta) = \begin{cases} \frac{e^{prefer_{a|s,\theta}}}{\sum_{x \in A_s} e^{prefer_{x|s,\theta}}} & a \in A_s \\ 0 & a \notin A_s \end{cases}$$

In this notation, $A_s$ stands for the available action space in state $s$.

#### 3.3.2. Priority Representation in FJSP

For the convenience of explaining, several symbols are defined:

K means the number of processing types.

$S_n = (s_{11}, s_{12}, \ldots, s_{1N}, s_2)$ means the scheduling state at node n.

$f_\theta()$ means the function that the NN represents, $\theta$ stands for the parameters of NN. In this article's method, there are $K + 1$ NNs, which are represented as $f_{\theta 11}, f_{\theta 12}, \ldots, f_{\theta 1K}, f_{\theta 2}$.

For the first kind of action, the priority is computed as follows:

$$\left( prefer_{a1ij_1|s,\theta}, prefer_{a1ij_2|s,\theta}, \cdots prefer_{a1ij_m|s,\theta} \right) = f_{\theta 1i}(s)$$

For the second kind of action, the priority is computed as follows:

$$prefer_{a2|s,\theta} = f_{\theta 2}(s)$$

#### 3.3.3. Objective Function

In the field of RL, objective function $J(\theta)$ is the quantity to measure the agent's performance and direct the agent to learn. Customarily, the objective function is the larger the better. For an episodic problem, the value function of the initial state is used to be the objective function. So, $J(\theta) = v_{\pi\theta}(s_0) \doteq E_{\pi\theta}\left[\sum_{k=0}^N \gamma^k r_{k+1}\right]$.

When $\gamma = 1$, for the reward is setup as $r_{n+1} = \begin{cases} 0 \\ -F_n & a_n \in A_1 \\ & a_n \in A_2 \end{cases}$, the objective function is $J(\theta) = E_{\pi\theta}\left[\sum_{k=0}^N r_{k+1}\right] = -E_{\pi\theta}[\Sigma F_n] = -E_{\pi\theta}[F]$. Thus, the target in MDP is $\max(-E_{\pi\theta}[F])$, which is consistent with the target min(F) in FJSP.

When $0 < \gamma < 1$, the target can be viewed as the negative idle time of the system.

#### 3.3.4. Update the Policy

In machine learning problems, the gradient-based idea is often used to update parameters. In this idea, parameters move towards their gradients in each step until reaching

the convergence. For example, gradient ascent is $\theta \leftarrow \theta + \partial \nabla_\theta v_{\pi\theta}(s_0)$. In this notation, $\partial$ means the learning rate. $\nabla_\theta v_{\pi\theta}(s_0)$ represents the gradient of $v_{\pi\theta}(s_0)$ at $\theta$.

Concretely, in the policy-based algorithm, the gradient of the objective function is calculated as follows: $\nabla_\theta v_{\pi\theta}(s_0) = (G_n - b(s_n)\nabla_\theta \ln(\pi(a_n|s_n,\theta)))$, in which $G_n = \sum_{k=n}^{N} \gamma^{k-n} r_{k+1}$ means the discounted reward and $b(s_n)$ represents a baseline to decrease the variance.

If the problem is episodic, the policy is only updated after a trial is finished. The whole computation procedure is as follows Algorithm 1:

---

**Algorithm 1:** The policy gradient with baseline.

---

Input: A differentiable policy $\pi(a|s,\theta)\pi(a|s,\theta)$
Input: A differentiable value function $\hat{v}(s,w)\hat{v}(s,w)$
Hyper parameters: learning rate $\partial^\theta > 0, \partial^w > 0$
Initialize policy parameter $\theta$ and value function parameter $w$
Loop (for each episode):
    Run an episode according to $\pi(\cdot|\cdot,\theta)$, gets $s_0, a_0, r_1, \ldots s_{N-1}, a_{N-1}, r_N$,
        Loop (for each step in the episode) $n = 0, 1, \ldots, N-1$:

$$G \leftarrow \sum_{k=n+1}^{N} \gamma^{k-n-1} r_k$$
$$\delta \leftarrow G - \hat{v}(s_n, w)$$
$$w \leftarrow w + \partial^w \gamma^n \delta \nabla_w \hat{v}(s_n, w)$$
$$\theta \leftarrow \theta + \partial^\theta \gamma^t \delta \nabla_\theta \ln(\pi(a_n | s_n, \theta))$$

---

### 3.3.5. Modes of Choosing Action

There are two different stages: the training stage and the evaluating stage, which have different modes of choosing action.

In the training stage, the main responsibility of the agent is to learn and adjust the policy. So, the agent chooses action in the random mode, $a_n \sim \pi(a | s_n, \theta)$.

The training procedure of the scheduling agent is shown in Figure 7.

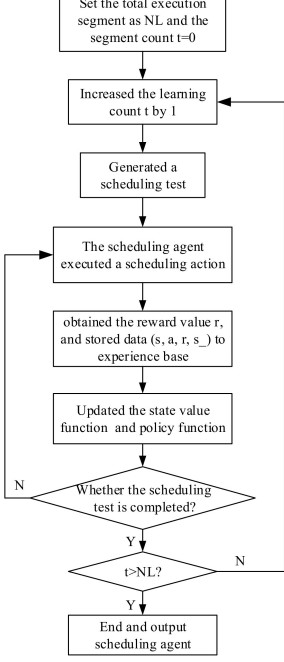

**Figure 7.** The training procedure of scheduling agent.

Step 1: Set the total execution segment as NL and the segment count t = 0 to control when the training ends.

Step 2: Increased the learning count by 1, t = t + 1.

Step 3: Generated a scheduling test.

Step 4: The scheduling agent executed a scheduling action, specifically a dispatching action or a processing action. The action mode followed by the agent during training is $a_n \sim \pi(a \mid s_n, \theta)$.

Step 5: The scheduling agent reached the next state, obtained the reward value r, and stored data (s, a, r, s_) to experience base.

Step 6: Updated the state value function $\hat{v}(s, w)$ and policy function $\pi(a|s, \theta)$ of the agent.

Step 7: Determined whether the current scheduling test was completed, if completed, continue. Otherwise, go to step 4.

Step 8: Judged whether the training ends according to whether it was greater than, and if it ends, output the scheduling agent; Otherwise, go to step 2.

In the evaluating stage, the responsibility of the agent is to obtain as large as reward as possible. So, the agent chooses action in the fixed mode; that is, $a_n = \underset{a}{\arg\max} \, \pi(a \mid s_n, \theta)$.

## 4. Simulation and Experiment

### 4.1. Simulation Environment

The simulation environment is an intelligent job-shop and there are two milling machines, two lathes, two carving machines, and warehouse in it. The layout of the intelligent job-shop is shown in Figure 8. The information type of these machines is shown in Table 2. For the development of the RL algorithm based on the MDP model, we choose Python 3.6 and tensorflow-gpu 1.2.0, which are a free development platform.

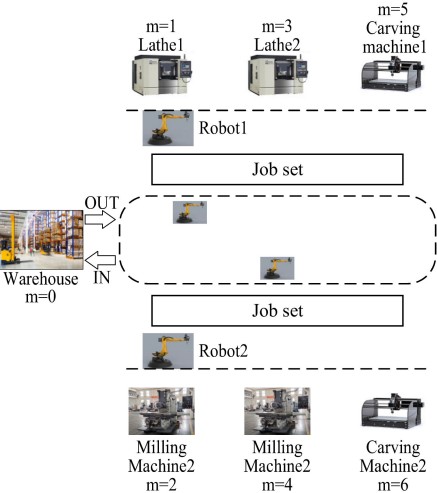

**Figure 8.** The layout of the simulated environment.

**Table 2.** The processing equipment information.

| Equipment Number | Equipment Type |
| :---: | :---: |
| M1 | Lathe |
| M2 | Lathe |
| M3 | Milling machine |
| M4 | Milling machine |
| M5 | Carving machine |
| M6 | Carving machine |

### 4.2. Experiment Results and Analysis

The experiment is mainly to verify the feasibility and effectiveness of the proposed real-time scheduling algorithm. A batch of 10 jobs is selected as the experimental example

(details are shown in Table 3). The number in the table is the processing time, and "-" means that the process cannot be processed on this machine.

**Table 3.** Job information.

| Jobs | Operations | Processing Time/s | | | | | |
|------|-----------|------|------|------|------|------|------|
| | | **M1** | **M2** | **M3** | **M4** | **M5** | **M6** |
| $J_1$ | $O_{11}$ | 9 | 7 | — | — | — | — |
| | $O_{12}$ | — | — | 11 | 9 | — | — |
| | $O_{13}$ | — | — | — | — | 4 | 5 |
| $J_2$ | $O_{21}$ | 8 | 10 | — | — | — | — |
| | $O_{22}$ | — | — | — | — | 5 | 3 |
| $J_3$ | $O_{31}$ | — | — | 12 | 10 | — | — |
| | $O_{32}$ | 8 | 6 | — | — | — | — |
| | $O_{33}$ | — | — | — | — | 5 | 7 |
| $J_4$ | $O_{41}$ | — | — | 5 | 8 | — | — |
| | $O_{42}$ | — | — | — | — | 7 | 9 |
| | $O_{43}$ | 7 | 9 | — | — | — | — |
| $J_5$ | $O_{51}$ | — | — | 6 | 8 | — | — |
| | $O_{52}$ | 9 | 5 | — | — | — | — |
| | $O_{53}$ | — | — | — | — | 2 | 3 |
| $J_6$ | $O_{61}$ | — | — | 6 | 4 | — | — |
| | $O_{62}$ | 10 | 12 | — | — | — | — |
| | $O_{63}$ | — | — | — | — | 7 | 5 |
| $J_7$ | $O_{71}$ | 10 | 8 | — | — | — | — |
| | $O_{72}$ | — | — | 6 | 8 | — | — |
| | $O_{73}$ | — | — | — | — | 5 | 7 |
| $J_8$ | $O_{81}$ | 13 | 11 | — | — | — | — |
| | $O_{82}$ | — | — | 3 | 5 | — | — |
| | $O_{83}$ | — | — | — | — | 6 | 8 |
| $J_9$ | $O_{91}$ | 9 | 6 | — | — | — | — |
| | $O_{92}$ | — | — | 8 | 7 | — | — |
| | $O_{93}$ | — | — | — | — | 8 | 7 |
| $J_{10}$ | $O_{101}$ | — | — | 3 | 8 | — | — |
| | $O_{102}$ | 9 | 5 | — | — | — | — |
| | $O_{103}$ | — | — | — | — | 5 | 8 |

The adaptive real-time scheduling method first learns 600 epochs and makespan per epoch is shown in Figure 9. It can be observed that by continuous learning, makespan is gradually reduced and finally stabilized between 45. The Gantt chart of the scheduling results by using the proposed adaptive scheduling method are shown in Figure 10. They meet the routing and machining constraints and prove the effectiveness of the proposed algorithm.

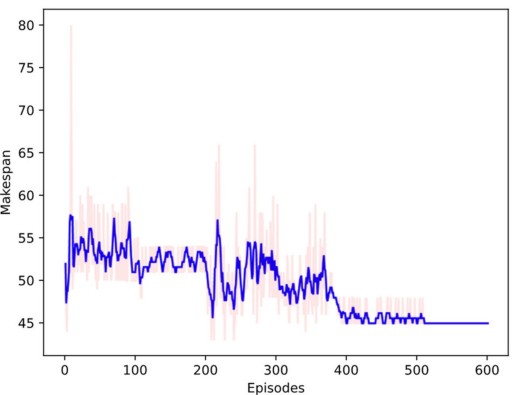

**Figure 9.** Makespan of each epoch.

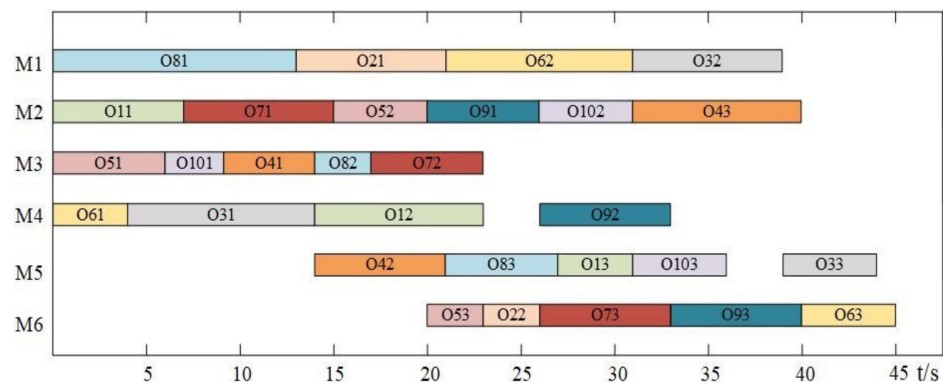

**Figure 10.** The Gantt chart of the scheduling results.

Some common heuristic scheduling rules can be divided into two categories: machine selection rules and buffer job sequencing rules. The machine selection rules are: Shortest Queue (SQ), Less Queued Element (LQE) and Shortest Processing Time (SPT). Buffer job sequencing rules are: First In, First Out (FIFO), Shortest Job First (SJF) and Last In, First Out (LIFO). Therefore, these six single rules can be combined into nine different rule sets to select machine and job: SQ + FIFO, SQ + SJF, LQE + FIFO, LQE + SJF. LQE + LIFO, SPT + FIFO, SPT + SJF, SPT + LIFO. Compared with the single traditional scheduling rule proposed above, the adaptive real-time scheduling method proposed in this paper can select the current optimal scheduling rule at each time by training agents to adapt to the environment. Therefore, in order to illustrate the effectiveness of the method in this paper, it is compared with the single rule method mentioned above, and the makespan of each method is obtained under the same other conditions. The results show that our method can achieve a better solution than the single dispatch rule method, as shown in Figure 11. The result of the proposed method has more than 10% improvement, even compared to the best rule SPT + FIFO of the 9 single dispatching rules on this simulation experiment.

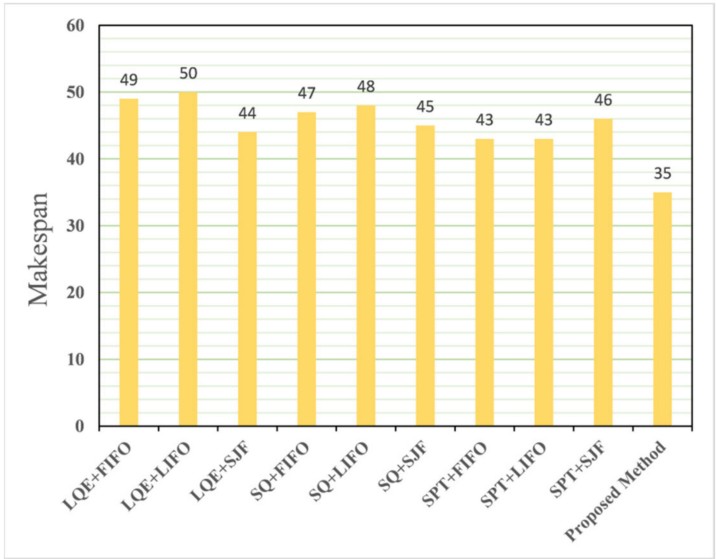

**Figure 11.** Comparison between various methods.

## 5. Conclusions

Aiming at the transformation of product information from single variety and large batch to multi-variety and small batch in enterprise job-shop, this paper proposes a real-time production scheduling method based on reinforcement learning to solve DFJSP effectively. After reviewing the traditional job-shop scheduling methods and the dynamic and static

scheduling methods of flexible job-shop, a flexible job-shop scheduling model based on reinforcement learning is established.

Considering the low efficiency of the traditional job-shop scheduling method, this paper first studies the principles of RL and finds the relationship between the MDP and flexible job-shop scheduling process. A manufacturing neural network that can input high-dimensional information of jobs and machines is proposed in this paper. Then, a policy-based reinforcement learning algorithm is proposed to achieve the optimum objective. Finally, the proposed methodology is evaluated and validated with experiments in a smart manufacturing setting. Future research will focus on solving the problem of uncertain delivery time, because it is a common phenomenon that the lead time of productions changes due to various factors. This paper makes an attempt to improve the AI for production scheduling. We hope this work will help catalyze more in-depth investigations and multi-disciplinary research efforts to advance the AI for smart manufacturing.

**Author Contributions:** Conceptualization, H.Z.; methodology, H.Z.; software, H.Z.; validation, S.T. and Y.G.; formal analysis, Q.C.; investigation, S.T.; writing—original draft preparation, H.Z.; writing—review and editing, S.T. and Y.G.; visualization, Q.C.; funding ac-quisition, H.Z. All authors have read and agreed to the published version of the manuscript.

**Funding:** This work was supported by National Key Research and Development Program of China No. 2018YFE0177000, and the Fundamental Research Funds for the Central Universities [No. NT2021021].

**Data Availability Statement:** Not applicable.

**Conflicts of Interest:** The authors declare no conflict of interest.

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
