# Peer review of "Research on an Adaptive Real-Time Scheduling Method of Dynamic Job-Shop Based on Reinforcement Learning"

_machines, doi:10.3390/machines10111078_

Round 1

Reviewer 1 Report

In this work, the authors propose the study of a FJSP using reinforcement learning, discussing their relevance as a method for adaptive real-time scheduling. One experiment of 3 stages with 2 parallel machines and 10 jobs addresses the feasibility and efficiency of the approach. Overall, the paper addresses a very interesting topic for the research community, but in my opinion the authors are advised to verify the following issues concerning paper contributions and reasoning, which makes me argue against the publication of the manuscript in its current state. Overall, the language and structure of the main paper sections are adequate, but the Methodology section formatting should be improved to guide the reader, and MDP is the common framework environment of RL so it’s rather strange to be highlighted as contribution (or clarify what is done different). Likewise, please also clarify the claim that a “novel manufacturing neural network is developed” by offering the adequate contradiction and scope of novelty compared to literature. But mostly, I believe the main paper contributions are not demonstrated by the Results (far insufficient with such simple case), since the paper is oblivious of the proposed “real-time scheduling” concept, misleading on the stated comparison with other “common rule-based scheduling methods” (stated in Abstract), and misguiding to demonstrate that the proposed approach, as stated in the Conclusions, “has a significant effect on improving the real-time scheduling efficiency of dynamic job-shop”. Without any of this, the work is utmost incomplete.

Reviewer 2 Report

It is globally a meritory work ; nevertheless I would propose some suggestions to improve the quality of the article:

1. It should include the meaning of all acronyms. For examples, next ones are never fully pointed in the text:

FJSP

DQN-based

AGV

2. Literature review should be enlarged. It goes too quickly on the state of art, showing a too short, single idea for only some of the authors.

3. Related to methodology, it should explain why this type of NN. 

4. In results, I would expect also some explanations about the extrange results that NN offers in some cases. That is, when inputs does not exactly match the trials, we could expect anomalous behaviors. Has this point analyzed in the study?

5. English is generally good, but some sentences sound rare, for example:

There two kinds of action spaces in the set of available actions.

It is possible to find a few typos , just a final review could be enough

Reviewer 3 Report

1. The last 2 sentences in passage 1 need citation.

2. The 2nd sentence in passage 2 needs citation.

3. The sentences "However, ..." in passage 2 needs citation.

4. The sentences "Dynamic ..." in passage 2 needs citation.

5. The sentences "Recently ..." in passage 3 needs citation.

6. The sentences "Based on the study ..." in passage 4 needs citation.

7. The sentences "But in the actual ..." in section 2.1 needs citation.

8. Why need to propose different method when RL can be the effective approach?

9. 2nd passage in section 3 needs citation.

10. Most of the sentences in 2nd passage of section 3.1.1 needs strong reasoning.

11. High dimentionality in section 3.2 needs strong reasoning.

12. Where did the 'New assigned operation' go? in the figure 6.

13. It strongly needs performance comparison on the obtained results to other similar works or different methods.  

Round 2

Reviewer 1 Report

In the previous review, the authors were advised to proceed to a major revision of their work, which I believe it was fairly neglected to overcome the major concerns raised in terms of contributions. In my view, the new text added in this version offers an even more confusing and misguiding reasoning (e.g. what is the meaning used for "artifact"?), and the answers provided were insufficient to demonstrate neither the "real-time scheduling efficiency", nor the novelty of the "novel manufacturing neural network" of the proposed approach. If neither of these claims can be concluded from the results discussed (which in resume just prove a learning convergence, and now with a comparison with other dispatching rules without adequate context or explanation of them), in conscience these cannot be statements present in the work. Therefore, in accordance with the guidelines, unfortunately the article pursues serious flaws to make me argue against its publication in its current form. 

Reviewer 3 Report

in overall the paper is sufficient for a publication after another carefull english proofread.

Round 3

Reviewer 1 Report

I acknowledge the effort of the authors in clarifying my concerns, I believe the answers provided are sufficient to make the paper suitable for publication.